# Spatial analysis of COVID-19 spread in Iran: Insights into geographical and structural transmission determinants at a province level

Ricardo Ramírez-Aldana[1]*, Juan Carlos Gomez-Verjan[1], Omar Yaxmehen Bello-Chavolla[1,2]

1 Research Division, Instituto Nacional de Geriatría, Mexico City, Mexico, 2 Department of Physiology, Facultad de Medicina, Universidad Nacional Autónoma de México, Mexico City, Mexico

* ricardoramirezaldana@gmail.com

**Data Availability Statement:** All relevant data are within the manuscript and its Supporting Information files.

## Abstract

The Islamic Republic of Iran reported its first COVID-19 cases by 19th February 2020, since then it has become one of the most affected countries, with more than 73,000 cases and 4,585 deaths to this date. Spatial modeling could be used to approach an understanding of structural and sociodemographic factors that have impacted COVID-19 spread at a province-level in Iran. Therefore, in the present paper, we developed a spatial statistical approach to describe how COVID-19 cases are spatially distributed and to identify significant spatial clusters of cases and how socioeconomic and climatic features of Iranian provinces might predict the number of cases. The analyses are applied to cumulative cases of the disease from February 19th to March 18th. They correspond to obtaining maps associated with quartiles for rates of COVID-19 cases smoothed through a Bayesian technique and relative risks, the calculation of global (Moran's I) and local indicators of spatial autocorrelation (LISA), both univariate and bivariate, to derive significant clustering, and the fit of a multivariate spatial lag model considering a set of variables potentially affecting the presence of the disease. We identified a cluster of provinces with significantly higher rates of COVID-19 cases around Tehran (p-value< 0.05), indicating that the COVID-19 spread within Iran was spatially correlated. Urbanized, highly connected provinces with older population structures and higher average temperatures were the most susceptible to present a higher number of COVID-19 cases (p-value < 0.05). Interestingly, literacy is a factor that is associated with a decrease in the number of cases (p-value < 0.05), which might be directly related to health literacy and compliance with public health measures. These features indicate that social distancing, protecting older adults, and vulnerable populations, as well as promoting health literacy, might be useful to reduce SARS-CoV-2 spread in Iran. One limitation of our analysis is that the most updated information we found concerning socioeconomic and climatic features is not for 2020, or even for a same year, so that the obtained associations should be interpreted with caution. Our approach could be applied to model COVID-19 outbreaks in other countries with similar characteristics or in case of an upturn in COVID-19 within Iran.

**Funding:** This project was supported by a grant from the Secretaría de Educación, Ciencia, Tecnología e Innovación de la Ciudad de México CM-SECTEI/041/2020 "Red Colaborativa de Investigación Traslacional para el Envejecimiento Saludable de la Ciudad de México (RECITES) (R.R. A). The funders had no role in study design, data collection and analysis, decision to publish, or preparation of the manuscript.

**Competing interests:** The authors have declared that no competing interests exist.

## Author summary

Iran was among the first countries reporting a rapid increase in the number of COVID-19 cases. Spatial epidemiology is useful to study the spatial distribution of a disease and to identify factors associated with the number of cases of such disease. By applying these methods, we aimed to identify whether there are clusters of regions in Iran with high or low number of COVID-19 cases and the association of different factors with these numbers, considering spatial relationships and maps representing these associations. Interestingly, we found regions of high number of cases and that more COVID-19 cases were present in provinces with more urbanization, aging population, number of physicians, efficient communications, and greater average temperatures, whereas less COVID-19 cases were present in provinces with more literacy. This study allowed us to understand the spatial behavior of the disease and the importance of having adequate health policies, literacy campaigns, and disseminating health information to the population.

## Introduction

On 11th March 2020, the General Director of the World Health Organization (WHO), Dr. Tedros Adhanom Ghebreyesus, declared the new infectious respiratory disease COVID-19, caused by the infection of novel coronavirus SARS-CoV-2 as a pandemic, due to the rate of growth of new cases, the number of affected people, and the number of deaths [1]. As of the time of this writing (April 15th, 2020), the number of infected cases world-wide corresponded to more than 1 million, being the most affected countries: Italy (16,523 deaths), Spain (13,341 deaths), USA (10,792 deaths), France (8,911 deaths), United Kingdom (5,373 deaths), and Iran (3,739 deaths) [2,3].

Iran was among the first countries outside of China to report a rapid increase in the number of COVID-19 cases and associated deaths; its first confirmed cases were reported on 19th February 2020 in the province of Qom imported from Wuhan, China [4]. Nevertheless, some reports suggest that the outbreak may have happened two or six weeks before the government official announcement [5]. Iran had one of the highest COVID-19 mortality rates early in the pandemic, and its rate of spread has been amongst the highest. However, as with other countries, it may be a sub-estimation of cases, and there may be other cases not officially reported [6].

The large count of COVID-19 cases and mortality in Iran are multifactorial. Iran's response to the epidemic has been highly affected by several imposed economic sanctions and armed conflicts within the last 20 years. Moreover, its difficult economic situation due to a recession, having inflation rates that are among the highest in the region, has taken a toll on its public health system [7,8]. Although there are approximately 184,000 hospitals and primary healthcare staff, limitations in the availability of COVID-19 testing kits, protective equipment, and ventilators are quite important. On the other hand, over the last years, Iran has slowed the rate of mortality associated with infectious and maternal diseases. It is currently undergoing an epidemiological transition where infectious diseases interact with chronic conditions [2]. In this sense, Iran may represent other similar developing world countries with poor health systems and an increased prevalence of chronic diseases.

Spatial statistics have emerged as a useful tool for the analysis of spatial epidemiology, concerning mapping and statistical analyses of spatial and spatio-temporal incidences of different pathogens. The aim of this paper is to perform spatial analyses which allow us to better understand the COVID-19 outbreak in Iran, not only in terms of the strength of its presence and

socioeconomic and structural factors which facilitate the disease spread within Iranian provinces, but also in terms of how the disease is spatially distributed and which variables are spatially related with it considering the spatial effect to obtain adequate inferences. Given the role of climate and socio-economic factors in determining the distribution of cases and its impact world-wide, we also aimed to incorporate said factors as predictors of SARS-CoV-2 spread [9,10]. This could aid to understand the burden of COVID-19, its distribution in the country, and its implication on public health within Iran and similar countries [11] and could contribute to public health measures by providing insight to inform the implementation of interventions or to understand socio-demographic factors associated with the SARS-CoV-2 spread and COVID-19 heterogeneity as it has been applied to previous infectious diseases [12–16].

## Material and methods

### Data sources

We obtained province-specific data considering 31 provinces or polygons in Iran (**Table 1**). From the Statistical Centre of Iran [17], we extracted information concerning: 1) people settled in urban areas in 2016 (%), calculated from the population and household of Iran by province and sub-province information of the census, 2) people aged $\geq$60 years in 2016 calculated from the population disaggregated by age groups, sex, and province information of the census, 3) population density (people per $km^2$) in 2016, 4) literacy rate of population aged $\geq$6 years in 2016, obtained from the document of selected results from the 2016 census, 5) the Consumer Price Index percent changes on March 2020 for the national households in contrast to the corresponding month of the previous year (point-to-point inflation), and 6) the average temperature (˚C) of provincial capitals and 7) annual precipitation levels (mm) in 2015, both part of the climate and environment information. From the Iran data portal [18], we obtained, from the health section: 1) the number of physicians employed by the ministry of health and medical education in 2006 and 2) the number of beds in operating medical establishments in 2006 and from the government finance section: 3) the province contribution to gross domestic product (GDP) in 2004. We used a Transportation Efficiency Index (TEI) [19], constructed through Data Evelopment Analysis, being an indicator of the extent in which each province efficiently utilize their transportation infrastructure. The TEI has values between zero and one, values near to one indicate provinces better communicated, but we standardized it (values of each province minus its mean divided by the associated standard deviation) to allow interpretations in a better scale in terms of how the increase in a certain number of standard deviations of the TEI is associated with the number of COVID-19 cases. The cumulative number of cases with confirmed COVID-19 by Province from February 19[th] to March 18[th], 2020, was also obtained [20]. It is important to notice that, in order to obtain more accurate rates of cases with COVID-19, population size in 2020 by province was derived by using mathematical projection methods (arithmetic, geometric, exponential, and logistic or saturation methods) using information contained in the population and housing censuses from 2006, 2011, and 2016. Since all methods provided similar results, we show here only those associated with the arithmetic method. Shapefiles were obtained from the Stanford Libraries Earthworks: https://earthworks.stanford.edu/catalog/stanford-dv126wm3595, in which files are freely available for academic use and other non-commercial use [21].

### COVID-19 rate estimation by Iranian provinces

We obtained quantile maps associated with raw rates of COVID-19 cases, as well as smoothed case rates by province using an empirical Bayes estimator, which is a biased estimator that improves variance instability proper of rates estimated in small-sized spatial units [22] (i.e.

**Table 1. Features extracted for spatial analyses disaggregated by Iranian provinces to predict the spread of COVID-19 cases.** Abbreviations: GDP, Gross Domestic Product; TEI, Transportation Efficiency Index

| Province | Cases[+] | Urban population (%)[*] | >60 years (%)[*] | Area (km²)[*] | Density[*] | Literacy (%)[*] | Average temperature (°C)[*] | Annual precipitation (mm)[*] | Physicians[**] | GDP[**] (2004) | Hospital beds[**] (2006) | Inflation[*] | TEI[***] | Population[****] (2020) |
|---|---|---|---|---|---|---|---|---|---|---|---|---|---|---|
| Alborz | 906 | 92.639 | 8.914 | 5122 | 529.559 | 92.2 | 16.7 | 220.5 | 2632.5 | 12.577 | 15327.5 | 28 | 0.524 | 2952309.6 |
| Ardebil | 213 | 68.169 | 9.371 | 17800 | 71.372 | 83.1 | 10.9 | 296.5 | 354 | 1.127 | 1654 | 23.4 | 0.46 | 1287965.6 |
| Bushehr | 46 | 71.854 | 6.841 | 22743 | 51.154 | 89.2 | 26.5 | 272.5 | 429 | 3.227 | 1345 | 24.4 | 0.301 | 1267760.8 |
| Chahar Mahall and Bakhtiari | 58 | 64.092 | 8.691 | 16328 | 58.045 | 84.7 | 11.8 | 309.7 | 499 | 0.727 | 1234 | 22.7 | 0.754 | 989763 |
| East Azarbaijan | 571 | 71.859 | 10.732 | 45651 | 85.642 | 84.7 | 14 | 286.9 | 1104 | 3.927 | 5964 | 21 | 0.56 | 4057677.6 |
| Esfahan | 1538 | 88.019 | 10.643 | 107018 | 47.850 | 89.9 | 17.7 | 96.3 | 2109 | 6.527 | 8261 | 24.2 | 0.696 | 5314080.4 |
| Fars | 386 | 70.119 | 9.456 | 122608 | 39.567 | 88.8 | 18.9 | 271.5 | 1661 | 4.527 | 7154 | 22.3 | 0.591 | 5054966.8 |
| Gilan | 924 | 63.343 | 13.250 | 14042 | 180.223 | 87.3 | 17.3 | 1388.3 | 1211 | 2.327 | 3716 | 24 | 0.472 | 2570553.6 |
| Golestan | 351 | 53.275 | 7.796 | 20367 | 91.757 | 86.1 | 18.8 | 477.8 | 998 | 1.527 | 1769 | 25.4 | 0.372 | 1942263 |
| Hamadan | 155 | 63.123 | 10.801 | 19368 | 89.748 | 85 | 13.1 | 215.7 | 688 | 1.627 | 3089 | 22.9 | 0.429 | 1722206.8 |
| Hormozgan | 124 | 54.707 | 6.046 | 70697 | 25.127 | 87.8 | 27.8 | 152.2 | 492 | 2.227 | 1686 | 32.1 | 1 | 1935000.6 |
| Ilam | 120 | 68.130 | 8.508 | 20133 | 28.816 | 84.9 | 18 | 842.4 | 145 | 0.827 | 875 | 27.9 | 1 | 598205.2 |
| Kerman | 127 | 58.728 | 7.811 | 180726 | 17.511 | 81.5 | 17.2 | 109.8 | 955 | 2.527 | 3325 | 25.7 | 0.47 | 3345302 |
| Kermanshah | 152 | 75.220 | 10.023 | 25009 | 78.069 | 85.4 | 16.5 | 512.8 | 755 | 1.627 | 2922 | 22.6 | 0.536 | 1958199.6 |
| Khuzestan | 359 | 75.453 | 7.052 | 64055 | 73.539 | 86.3 | 27.3 | 269.7 | 1599 | 14.627 | 7511 | 22.3 | 0.95 | 4853540.2 |
| Kohgiluyeh and Buyer Ahmad | 45 | 55.741 | 7.139 | 15504 | 45.991 | 84.4 | 15.7 | 611.1 | 232 | 4.027 | 573 | 24.1 | 1 | 756590.4 |
| Kordestan | 189 | 70.756 | 9.304 | 29137 | 55.016 | 84.5 | 15.4 | 444.4 | 605 | 1.127 | 2155 | 18.8 | 0.818 | 1690503.8 |
| Lorestan | 363 | 64.460 | 8.830 | 28294 | 62.227 | 83 | 17.9 | 535.6 | 616 | 1.327 | 2153 | 26.7 | 0.963 | 1765773.8 |
| Markazi | 782 | 76.935 | 10.892 | 29127 | 49.077 | 87 | 15.1 | 284.8 | 514 | 2.327 | 1866 | 23.4 | 0.689 | 1441887.8 |
| Mazandaran | 1494 | 57.780 | 11.414 | 23842 | 137.723 | 88.7 | 18.6 | 724.7 | 1585 | 3.527 | 4475 | 25.9 | 0.269 | 3451293.2 |
| North Khorasan | 100 | 56.118 | 8.500 | 28434 | 30.354 | 83.3 | 14.8 | 227.4 | 288 | 0.727 | 730 | 24.7 | 0.533 | 859384 |
| Qazvin | 526 | 74.751 | 8.925 | 15567 | 81.824 | 88.6 | 15.7 | 313.7 | 429 | 1.427 | 1403 | 25.4 | 0.544 | 1331517.8 |
| Qom | 1074 | 95.178 | 7.696 | 11526 | 112.119 | 88.7 | 19.6 | 111.6 | 319 | 1.127 | 1493 | 24.6 | 1 | 1404771.8 |
| Razavi Khorasan | 661 | 73.058 | 8.478 | 118851 | 54.139 | 89.1 | 17.2 | 183.4 | 3328 | 5.027 | 9131 | 20.5 | 0.658 | 6786580.2 |
| Semnan | 577 | 79.803 | 9.978 | 97491 | 7.204 | 91.5 | 19.5 | 107.5 | 493 | 0.927 | 1269 | 22.7 | 0.868 | 759273.6 |
| Sistan and Baluchestan | 88 | 48.491 | 4.886 | 181785 | 15.265 | 76 | 19.8 | 103.7 | 657 | 1.127 | 2117 | 26.5 | 1 | 2967563.6 |
| South Khorasan | 100 | 59.023 | 9.757 | 95385 | 8.061 | 86.8 | 17.4 | 144.3 | 512 | 0.527 | 660 | 24.5 | 0.605 | 853989.2 |
| Tehran | 4260 | 93.854 | 10.443 | 13692 | 969.007 | 92.9 | 19.1 | 209.3 | 2632.5 | 12.577 | 15327.5 | 28 | 1 | 14135033.8 |
| West Azarbaijan | 300 | 65.423 | 8.562 | 37411 | 87.280 | 82 | 12.5 | 277.3 | 993 | 2.027 | 3630 | 23.3 | 0.644 | 3412933.4 |
| Yazd | 471 | 85.316 | 8.788 | 129285 | 8.806 | 90.9 | 21.3 | 38.4 | 610 | 1.227 | 2395 | 23.1 | 0.941 | 1189817 |
| Zanjan | 261 | 67.253 | 9.783 | 21773 | 48.568 | 84.8 | 14 | 283.1 | 492 | 1.027 | 1264 | 22.4 | 0.651 | 1090842.6 |

* Statistical Centre of Iran

** Iran data portal

*** Obtained from reference (12)

**** Population projected by using the population census 2011 and 2016 and an arithmetic method

[+]Cases obtained from John Hopkins Database (https://coronavirus.jhu.edu/map.html)

provinces with a larger population size have lower variance than provinces with a smaller population size). Since raw and smoothed rates were surprisingly similar, only results of smoothed rates are reported. We also obtained maps concerning excess or relative risk, serving as a comparison of the observed number of cases by the province to a national standard. For the variable concerning the number of people aged ≥60 years by province, raw, and smoothed rates were obtained using empirical Bayes, and the latter were used in all analyses.

## Spatial weight estimation and spatial autocorrelation

Since all spatial analyses require spatial weights, we obtained queen contiguity weights [23]. Provinces were considered as neighbors when they share at least a point or vertex in common,

obtaining a squared matrix of dimension 31 (31x31 matrix) with all entries equal to zero or one, the latter value indicating that two provinces are neighbors. From these neighbors, weights are calculated by integrating a matrix in a row-standardized form, i.e., equal weights for each neighbor and summing one for each row. Moran's I statistic was obtained as an indicator of global spatial autocorrelation [24], and its significance was assessed through a random permutation inference technique based on randomly permuting the observed values over the spatial units [25]. Local indicators of spatial autocorrelation (LISA) were obtained, being these a decomposition of Moran's I used to identify the contribution of each province in the statistic [26]. LISA was used to derive significant spatial clustering through four cluster types: High-High, Low-Low, High-Low, and Low-High. For instance, the High-High cluster indicates provinces with high values of a variable that are significantly surrounded by regions with similarly high values. Analogous to Moran's I and LISA, estimates can be calculated to identify the spatial correlation between two variables and to identify bivariate clustering [27]. For instance, to identify provinces with high values in a first variable surrounded by provinces with high values for a second variable (cluster High-High). Bivariate clustering and quartile maps were obtained for each of the significant variables in a linear spatial model, to have a better understanding of the individual spatial effect of each of these variables over the smoothed rates associated with the disease.

## Spatial multivariate linear models

Spatial multivariate linear models were fitted to identify variables that significantly impact the number of log-transformed COVID-19 cases [28]. This response variable was chosen since the corresponding model better satisfies all statistical assumptions, the other variables introduced in the Data section were simultaneously introduced as explanatory, first removing from the model all variables generating multicollinearity. Ordinary Least Squares (OLS) estimation was used to identify whether a linear spatial model was necessary by using a Lagrange Multiplier (LM) and a robust LM statistics to compare the non-spatial model with spatial models [29]. Two kinds of spatial models were compared; the spatial-lag model considers the spatially lagged response as an additional explanatory variable, whereas the spatial-error model considers that the error is a linear function of a spatially lagged error plus another error term. Another model was obtained by performing a backward selection process, considering the elimination of the most non-significant variable in each step and the minimization of the Akaike Information Criterion (AIC). This process allowed us to identify whether the associations obtained through this model were similar as those obtained through the model including all variables. For significant variables in the linear spatial models, interpretations in the original scale (i.e., as counts) were derived and we performed bivariate LISA significant clustering between each of these significant variables and the rate of cases with COVID-19, as explained above. All statistical analyses were conducted using GeoDa version 1.14.0. A two-tailed p-value<0.05 was considered as the significance threshold.

## Results

### Rates description and spatial autocorrelation of COVID-19 case rates between provinces

Maps for quartiles corresponding to the smoothed rates and excess risk of COVID-19 cases are shown in **Fig 1**. We observed that the highest rates of COVID-19 and excess risk values were located in the Northern region of Iran corresponding to the provinces of Qom, Marzaki, Mazandaran, and Semnan. There were also high rates associated with the provinces of Alborz,

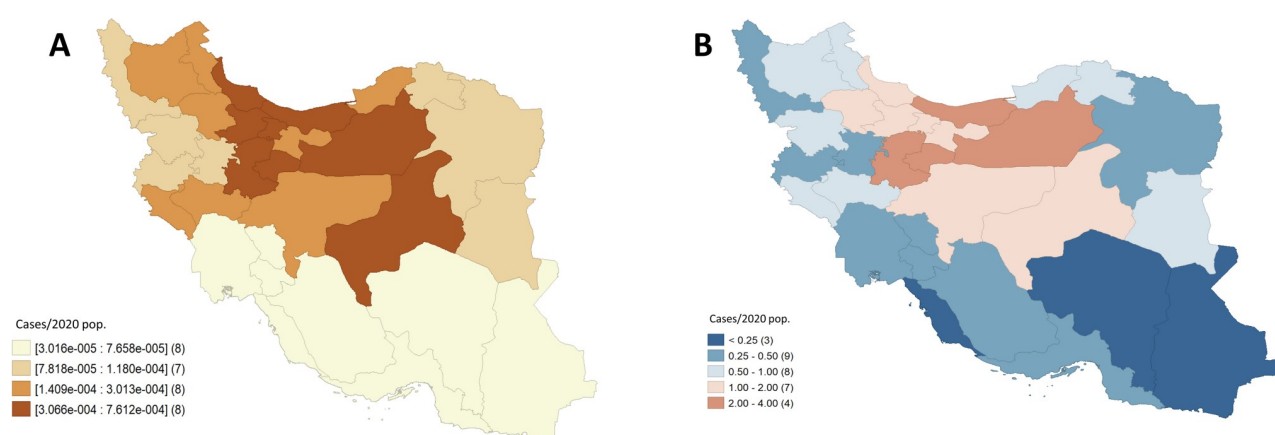

**Fig 1. Maps associated with COVID-19 cases by province from February 19th to March 20th, 2020.** A) Quartiles corresponding to rates smoothed through an empirical Bayes procedure. B) Excess or relative risk. Shapefiles obtained from the Stanford Libraries Earthworks: https://earthworks. stanford.edu/catalog/stanford-dv126wm3595.

Gilan, Qazvin, and Yazd (last quartile). We observed significant spatial autocorrelation (Moran's I = 0.426, p = 0.002), indicating that COVID-19 rates between provinces are significantly spatially related. From the heat and significance maps corresponding to significant clusters using an empirical Bayes spatial technique, we delimited a High-High cluster in red, indicating a northern zone around Tehran and Alborz with significant high COVID-19 rates surrounded by areas with similarly high rates. Conversely, we delimited a Low-Low cluster in blue indicating southern provinces with small rates surrounded by areas with similarly lower rates, which includes the provinces of Bushehr, Homozgan, Sistan, and Baluschestan. Interestingly, Golestan showed in light purple, has significantly lower COVID-19 rates despite being surrounded by a cluster of provinces with higher rates (**Fig 2**).

## Selection of multivariate linear spatial model for COVID-19 spread

Since the variable hospital beds is strongly associated with variables GDP and number of physicians (Kendall correlation coefficients above 0.55), we eliminated it from all models to avoid multicollinearity (S1 Fig). We confirmed that a spatial model was necessary since the error term from the OLS fitting showed significant spatial autocorrelation (Moran's I = 0.134, p-value = 0.025). Additionally, the LM and Robust LM statistics indicated that a spatial lag model was required since the spatial parameter ($\rho$) associated with the spatially lagged response was significant (LM = 10.669, p-value = 0.001; Robust LM = 13.557, p-value < 0.001), which did not occur with the spatial error model since the corresponding spatial parameter was not consistently significant (LM = 1.256, p-value = 0.262; Robust LM = 4.144, p-value = 0.042). Thus, only the spatial lag model was fitted obtaining a significant spatial parameter ($\rho$ = 0.723, p-value<0.001), which indicated that the rate of an area in the linear model is affected by COVID-19 rates in neighboring areas (R$^2$ = 0.877, $\sigma^2$ = 0.146). Normality and homoscedasticity assumptions were reasonably satisfied.

## Predictors of COVID-19 spatial spread in Iranian provinces

The significant variables associated with the model obtained through the selection scheme were the same as those associated with the model including all variables. This simplified model excluded population density, Consumer Price Index, and annual precipitation. The estimated coefficients were similar for both models; however, we analyzed the estimations associated

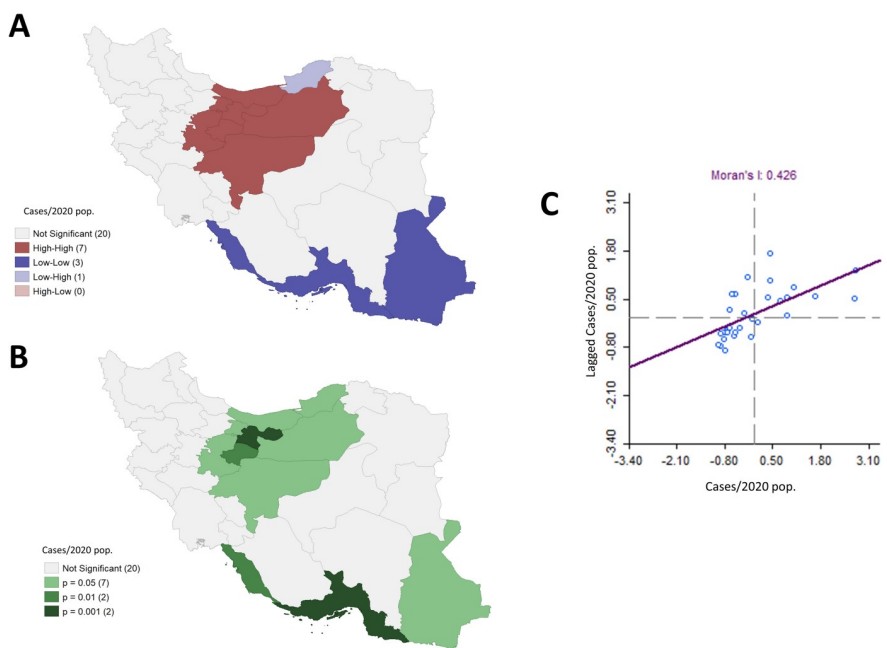

**Fig 2. Spatial clustering associated with rates of COVID-19 cases by province from February 19th to March 20th, considering queen contiguity weights.** A) Significant spatial clustering obtained through Local Indicators of Spatial Autocorrelation (LISA) comparisons. Four types of a cluster are possible: High-High, Low-Low, High-Low, and Low-High. For instance, the High-High cluster (red) indicates provinces with high values of a variable that are significantly surrounded by regions with similarly high values. B) P-values associated with the spatial clustering in A), C) Scatter plot associated with the smoothed rates vs. their corresponding spatially lagged values, including the associated linear regression fitting, whose slope corresponds to the Moran's I statistic, a global spatial autocorrelation measure. Shapefiles obtained from the Stanford Libraries Earthworks: https://earthworks.stanford.edu/catalog/stanford-dv126wm3595.

with the model including all variables to consider effects controlled for these three variables. Hence, the variables that significantly impact the log-transformed number of COVID-19 cases include: the percentage of people settled in urban areas (p-value = 0.019), smoothed rate of people aged ≥60 years (p-value < 0.001), literacy rate (p-value = 0.006), average temperature (p-value < 0.001), number of physicians employed (p-value < 0.001), and the TEI (p-value = 0.035) (**Table 2**). A 10% increase in urban population or a 1% increase in the population aged ≥60 years has a percentage increase of 29.29% (95%CI 26.55–32.10%) and 46.65% (95%CI 26.54% -69.95%), respectively, on the number of COVID-19 cases. Moreover, an increase of 1˚C in the temperature levels, an increase of 1 physician, or an increase of one deviation over the standardized TEI also have a percentage increase of 11.98% (95%CI 5.54–18.80%), 0.08% (95%CI 0.06–0.11%), and 16.98% (95%CI 1.14–35.30%), respectively, over the number of COVID-19 cases. Finally, a 1% increase in the literacy rate showed a percentage decrease of 10.44% (95%CI 3.18–17.16%) on the number of cases.

## Spatial lag predictors and province clusters

Quartile maps for each of the significant variables in the spatial lag model are shown in **Fig 3**. Finally, concerning bivariate LISA significant clustering (**Fig 4**), we observed a positive spatial relationship (Moran's I = 0.341, p-value = 0.002) between urban population and COVID-19 rates; provinces with high urban rates surrounded by areas with high COVID-19 rates are the same as the ones in the High-High cluster for COVID-19, except for Mazandaran, and similarly for the Low-Low cluster, except for Bushehr. There is also a positive spatial relationship

**Table 2. Spatial lag models estimated via maximum likelihood to predict log-transformed COVID-19 case distribution between Iranian provinces (model including all variables and model obtained through a selection scheme).**

| Variable | Coefficient | | SE | | z-value | | p-value | |
|---|---|---|---|---|---|---|---|---|
| | All variables[*] | Selection scheme[**] | All variables | Selection scheme | All variables | Selection scheme | All variables | Selection scheme |
| Spatial parameter (ρ) | 0.723 | 0.737 | 0.107 | 0.104 | 6.734 | 7.069 | <0.001 | <0.001 |
| Model constant | 2.510 | 2.853 | 2.550 | 2.425 | 0.984 | 1.176 | 0.325 | 0.239 |
| Urban population (%) | 0.026 | 0.026 | 0.011 | 0.010 | 2.345 | 2.653 | 0.019 | 0.008 |
| Population aged ≥60 | 0.383 | 0.331 | 0.075 | 0.062 | 5.089 | 5.324 | <0.001 | <0.001 |
| Population density | -0.0002 | | 0.0007 | | -0.258 | | 0.797 | |
| Literacy | -0.110 | -0.103 | 0.040 | 0.040 | -2.771 | -2.591 | 0.006 | 0.010 |
| Average temperature (˚C) | 0.113 | 0.114 | 0.030 | 0.028 | 3.748 | 4.105 | <0.001 | <0.001 |
| Precipitation levels (mm) | -0.0003 | | 0.0003 | | -0.98 | | 0.327 | |
| Physician distribution | 0.0008 | 0.0008 | 0.0001 | 0.0001 | 5.740 | 5.746 | <0.001 | <0.001 |
| GDP | -0.051 | -0.057 | 0.038 | 0.034 | -1.343 | -1.699 | 0.179 | 0.089 |
| Consumer Price Index | 0.032 | | 0.039 | | 0.812 | | 0.417 | |
| TEI | 0.157 | 0.155 | 0.074 | 0.074 | 2.112 | 2.081 | 0.035 | 0.038 |

[*]**Likelihood Ratio Test = 15.628, p< 0.001 (no spatial vs spatial model); R2 = 0.877; AIC = 57.165; $\sigma^2$ = 0.146**

[**] **Likelihood Ratio Test = 18.682, p< 0.001 (no spatial vs spatial model); R2 = 0.872; AIC = 52.704; $\sigma^2$ = 0.152**

(Moran's I = 0.279, p-value = 0.002) between the population aged ≥60 and COVID-19 rates. Both High-High and Low-Low clusters include similar provinces as the ones in the clusters for COVID-19, except for Qom and Alborz, which have significantly lower rates of people aged ≥60 years but are spatially surrounded by areas with high disease rates. Concerning literacy rates, we also identified a positive spatial relationship between literacy and disease rates (Moran's I = 0.362, p-value = 0.005). The associated High-High cluster and that obtained for COVID-19 rates are formed by the same provinces, whereas in the south, Hormozgan and Bushehr have high literacy rates but are surrounded by areas with low disease rates.

Concerning average temperature levels, the global spatial autocorrelation is negative (Moran's I = -0.107, p-value = 0.103). The High-High clusters for temperature and COVID-19 rates are similar, except for Marzaki and Alborz, where there is significantly lower temperature surrounded by areas with high COVID-19 rates. In the south, there is a significantly high temperature with spatially lower disease rates.

There is a positive spatial relationship (Moran's I = 0.302, p-value = 0.003) between the number of physicians and the COVID-19 rate. There is a High-High cluster in the north with a High-Low zone between formed by Marzaki, Qom, and Semnan, with a significantly lower number of physicians; however, they are spatially surrounded by areas with higher disease rates. Concerning the TEI, the spatial correlation is close to zero (Moran's I = -0.096, p-value = 0.112), indicating a particular random global spatial relationship between TEI and the disease rate. The High-High cluster is the same as the High-High cluster for the disease, except for Mazandaran and Alborz, which have significantly low TEI but are surrounded by areas

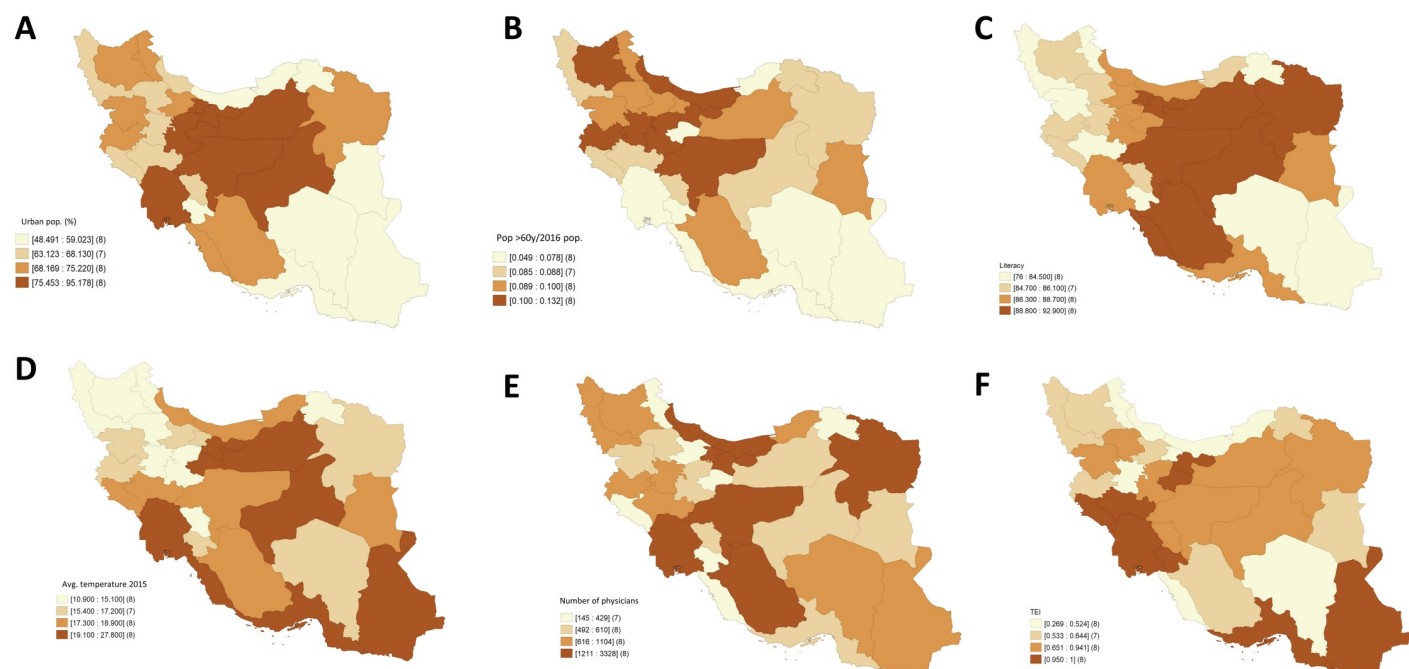

**Fig 3. Quartiles associated with all the explanatory variables significant in the spatial lag model with response variable the logarithm of the number of COVID-19 cases.** A) People settled in urban areas in 2016 (%). B) People aged ≥60 years, rates obtained through empirical Bayes smoothing. C) Literacy of population aged ≥6 years in 2016 (%). D) Average temperature (˚C) of provincial capitals in 2015. E) Number of physicians employed by the ministry of health and medical education in 2006. F) Transportation Efficiency Index (TEI). Shapefiles obtained from the Stanford Libraries Earthworks: https://earthworks.stanford.edu/catalog/stanford-dv126wm3595.

with high disease rates. In the south, two provinces, which formed a Low-Low cluster for COVID-19 cases, are now areas with high TEI spatially associated with areas with low disease rates.

## Discussion

Here, we demonstrate that the rates of COVID-19 cases within Iranian provinces are spatially correlated. This could be due to the origin of the outbreak, which started on the north of Iran, and can be seen through an important province cluster with the highest number of COVID-19 cases that we found around Tehran and Qom. Several mathematical models have been used to model the COVID-19 outbreak, mostly focused on forecasting the number of cases and assessing the capacity of country-level healthcare systems to manage disease burden [30–32]. In the present report, we demonstrate that the spatial relationship and socio-demographic factors associated with the provinces must be considered to model the disease adequately, and this report also highlights structural factors that may lead to inequities in COVID-19 spread. Of relevance, we highlight the role of social determinants of health in sustaining SARS-CoV-2 transmission and provide additional evidence that human mobility or province interconnectedness might be associated in favoring disease spread [33].

Importantly, our approach demonstrates that urbanization, aging population, education, average temperatures, number of physicians, and inter-province communications are associated with the case numbers amongst Iranian provinces. The obtained results do not consider the spatial effect, which is accumulated since the spatially lagged response is part of the explanatory variables, and they consider fixed values for all variables except the one being interpreted. Overall, these variables spatially correlate with the COVID-19 province clustering indicating a

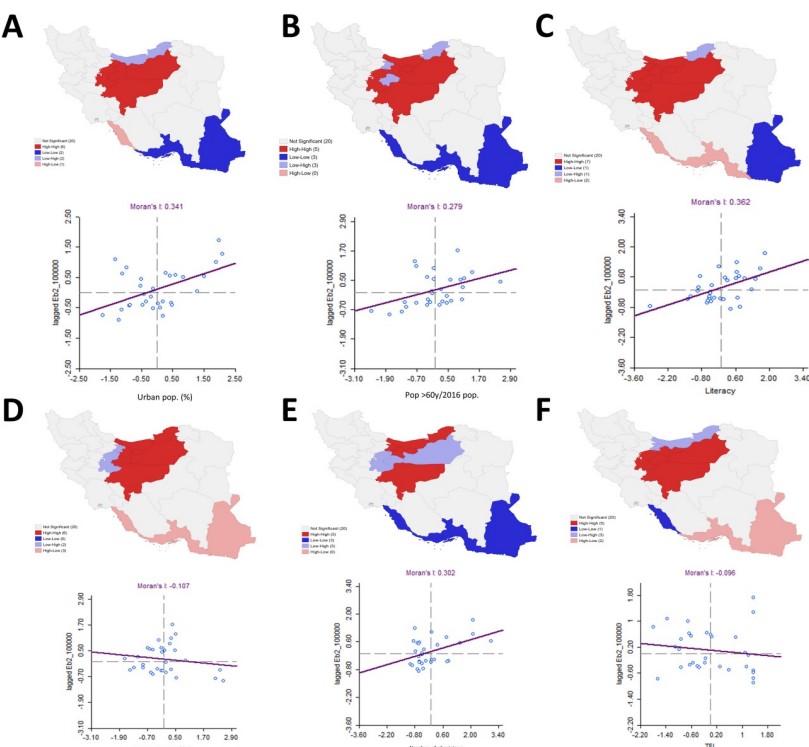

**Fig 4. Bivariate LISA's significant spatial clustering between each of the significant variables in the spatial lag model and the rate of cases with COVID-19 smoothed through the empirical Bayes approach.** The scatter plots associated with the variables vs. the spatially lagged smoothed rate of COVID-19 cases are presented as well, including the associated linear regression fitting, whose slope corresponds to the bivariate Moran's I statistic, a global spatial bivariate autocorrelation measure. A) People settled in urban areas in 2016 (%). B) People aged ≥60 years, rates obtained through empirical Bayes smoothing. C) Literacy of population aged ≥6 years in 2016 (%). D) Average temperature (˚C) of provincial capitals in 2015. E) Number of physicians employed by the ministry of health and medical education in 2006. F) Transportation Efficiency Index (TEI). Shapefiles obtained from the Stanford Libraries Earthworks: https://earthworks.stanford.edu/catalog/stanford-dv126wm3595.

consistent association with the observed variables. The greatest increase in the number of COVID-19 cases is associated with people aged ≥60 years, urban population, and how well the provinces are communicated, with age having one of the most important associations, an increase of 1% in the corresponding rate implies a percentage increase of 46.65% over the number of cases. Of relevance, mortality attributable to COVID-19 complications is higher in this age group, and age increases the likelihood of developing the symptomatic disease and increased disease severity [34,35]. Nevertheless, the association with older age could have different meanings depending on the number of comorbidities, with some reports labeling COVID-19 as an age-related disease [36]. Our data demonstrate that the spatial spread of COVID-19 has a relationship with population aging structures, a concept that must be explored in this setting to obtain population-specific estimates and lethality and which could represent a significant structural inequality related to COVID-19 burden [37].

Urbanization rates also are associated with a percentage increase over the number of COVID-19 cases; we observed a similar association regarding province interconnectedness, which goes in line with recent information on human mobility and its effect in decreasing disease spread through social distancing [33]. Urbanization, as a demographic phenomenon, leads to increased interconnectedness and human mobility as well as increased population density; these two factors facilitate disease spread. Emerging zoonotic diseases similar to

SARS-CoV-2 have been linked to major structural factors that have been reported in other studies, including population growth, climate change, urbanization, and pollution [38,39]. Thus, communication and the degree of urbanity, and what this implies in terms of pollution, overcrowding, among other factors, seem to be relevant to determine the number of COVID-19 cases and should imply geographical targets for public health interventions to monitor disease spread and disease containment [40].

The only effect associated with a decrease in the number of COVID19 cases in our study was attributed to literacy, which might reflect several factors that ultimately influence disease spread. Data from several countries, including Iran, identified that higher health literacy was associated with a lower number of COVID-19 cases, probably reflecting attitudes towards public health measures including social distancing, early disease detection, and hand hygiene [41,42]. Interestingly, this poses a potential public health intervention given that individuals with reduced health literacy, not only might have higher rates of COVID-19, but also increased likelihood for depression and impaired quality of life in suspected cases. Literacy's protective effect on disease spread also indicates a strong influence on social inequity and vulnerability as risk factors for COVID-19 spread, particularly on the influence of health equity, which will likely define the long-term impact of COVID-19 in many developing countries [43].

Concerning average temperature levels, we were able to obtain information associated only with the capitals and not the provinces, being a limitation of the analyzed information, obtaining some inconclusive results. On one hand, the global spatial autocorrelation was negative, though not statistically significant, indicating that global areas with higher temperatures are spatially related to areas with lower disease rates. On the other hand, on the spatial linear model, we derived that more temperature is associated with more cases. However; the former result does not contradict the latter since the direct effect in each province of a variable over the response is different from the spatial relationship between two variables. The latter considers one of the variables as spatially lagged (COVID-19), and thus the direct effect between variables in the same province is not included. In fact, this problem occurs in all the bivariate analysis, so care should be taken in all the interpretations. Notably, our results are consistent with previous analyses which have analyzed the impact of climate on SARS-CoV-2 stability and spread [44]. However, these results should be further studied considering the climate data limitations, that we obtained mixed results, and that some studies suggest there is no evidence that spread rates of the disease decline with higher temperatures [45].

Our study had some strengths and limitations. We approached COVID-19 using spatial analysis, which allowed us to identify province-level factors that are associated with the disease spread and which may be shared by other countries with similar socioeconomic or geographic structures by potentially identifying targets for country-wide public health interventions. This approach considers disease spread beyond individual-specific factors and could also be used to monitor areas of a potentially high number of undiagnosed cases that could facilitate disease spread and the surge of delayed waves of COVID-19 after initial mitigation [46,47]. Methodologically, all our analyses consider the spatial nature of the data. We identified significant spatial clustering and in terms of the spatial multivariate linear model, by including a spatial effect, we consider that the number of cases in an area is affected by those in neighboring areas. In this way, a lack of independency between spatial units is considered, being independence assumed in a usual linear model, thus obtaining more precise estimations. Of course, other statistical methods are available for this task, as generalized linear mixed models or geographically weighted regression; however, they do not use spatial weights, making our results more comparable with the Moran's I or spatial clustering, which are based on such weights. A limitation of our approach is that most of the variables used to explain COVID-19 disease rates were taken from previous years and not updates, given the unavailability of recent

estimates. Furthermore, smoothed COVID-19 rates were calculated using a projection of the population in 2020 since the most recent census corresponds to 2016, thus rates could have slightly different values. In this sense, the explanatory variables were not projected since information of previous years was not always available; however, precise projections for each variable were out of the scope of this work; and, it is also possible that some variables have a time lagged effect over the response. However; the time lagged effect we included was unintentional and dependent on the information available and not considering a lagged time effect as defined by experts; for instance, for GDP we used a time lag of 16 years, when perhaps it should have been of fewer years. When obtaining estimates using both projected and population size in 2016, we observed no significant changes in the results, which confirms the robustness of our approach. In fact, with all the mathematical projection methods similar results were obtained. However, the projections by province could be improved by considering a demographic balance equation and probabilistic projection methods as the ones obtained by country by the UN [48]. In this sense, we suspect similar results would still be obtained since our projected values by country are similar to those obtained by the UN. We also observed that the smoothed and raw rates of COVID-19 cases were similar, with an absolute difference between them of at most 0.607 (considering rates for every 1000 individuals), this was probably due to Iran not having provinces with extremely small or large population size. Future work could be focused on evaluating spatio-temporal modeling, which could be useful to monitor disease spread and identify additional factors relating not only to transmission rates but also to transmission dynamics. Since COVID-19 is currently challenging health systems all over the world, science-centered public health decisions could benefit from spatial modeling to investigate larger factors targeted for public health interventions.

In conclusion, COVID-19 spread within Iranian provinces is spatially correlated. The main factors associated with a high number of cases are older age, high degrees of urbanization, province interconnectedness, higher average temperatures, lower literacy rates, and the number of physicians. Structural determinants for the spread of emerging zoonotic diseases, including SARS-CoV-2, must be understood in order to implement evidence-based regional public health policies aimed at improving mitigation policies and diminishing the likelihood of disease re-emergence.

## Supporting information

**S1 Fig. Kendall correlation coefficients between all potential explanatory variables obtained to identify multicollinearity.**
(PNG)

## Author Contributions

**Conceptualization:** Ricardo Ramírez-Aldana, Juan Carlos Gomez-Verjan, Omar Yaxmehen Bello-Chavolla.

**Data curation:** Ricardo Ramírez-Aldana.

**Formal analysis:** Ricardo Ramírez-Aldana.

**Investigation:** Ricardo Ramírez-Aldana, Juan Carlos Gomez-Verjan, Omar Yaxmehen Bello-Chavolla.

**Methodology:** Ricardo Ramírez-Aldana.

**Project administration:** Ricardo Ramírez-Aldana.

**Software:** Ricardo Ramírez-Aldana.

**Supervision:** Ricardo Ramírez-Aldana.

**Validation:** Ricardo Ramírez-Aldana, Juan Carlos Gomez-Verjan, Omar Yaxmehen Bello-Chavolla.

**Visualization:** Ricardo Ramírez-Aldana, Omar Yaxmehen Bello-Chavolla.

**Writing – original draft:** Ricardo Ramírez-Aldana, Juan Carlos Gomez-Verjan, Omar Yaxmehen Bello-Chavolla.

**Writing – review & editing:** Ricardo Ramírez-Aldana, Juan Carlos Gomez-Verjan, Omar Yaxmehen Bello-Chavolla.

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
