## [Decision Letter · Decision Letter 0]

4 Jun 2020

Dear Dr. Ramírez-Aldana,

Thank you very much for submitting your manuscript "Spatial analysis of COVID-19 spread in Iran: Insights into geographical and structural transmission determinants at a province level" for consideration at PLOS Neglected Tropical Diseases. As with all papers reviewed by the journal, your manuscript was reviewed by members of the editorial board and by several independent reviewers. In light of the reviews (below this email), we would like to invite the resubmission of a significantly-revised version that takes into account the reviewers' comments. 

Reviews were obtained by experts in spatial statistics and/or spatial epidemiology, and while all three generally liked the analysis, reviewers 1 and 2 had some rather serious concerns either about the methods, or about the description of the methods. The authors are encouraged to substantially rework their methods according to these reviews. We cannot make any decision about publication until we have seen the revised manuscript and your response to the reviewers' comments. Your revised manuscript is also likely to be sent to reviewers for further evaluation.

Sincerely,

Townsend Peterson

Guest Editor

Justin Remais

Deputy Editor

I secured three reviews, each from an individual familiar with spatial statistics and/or spatial epidemiology. All three generally liked the analysis, but reviewers 1 and 2 had some rather serious concerns either about the methods, or about the description of the methods. I would like to see the authors rework their methods description rather deeply, and I will plan to have those two reviewers take a second look at the revised manuscript to assure that they are comfortable with the new descriptions.

Reviewer's Responses to Questions

**Key Review Criteria Required for Acceptance?**

**Methods**

-Are the objectives of the study clearly articulated with a clear testable hypothesis stated?

-Is the study design appropriate to address the stated objectives?

-Is the population clearly described and appropriate for the hypothesis being tested?

-Is the sample size sufficient to ensure adequate power to address the hypothesis being tested?

-Were correct statistical analysis used to support conclusions?

-Are there concerns about ethical or regulatory requirements being met?

Reviewer #1: "... Transportation Efficiency Index (TEI)[12], with values between zero and one, where values near to one indicate provinces better communicated, and considering such scale; we decided to standardize the TEI." The justification and the methods used are both rather unclear here.

"It is important to notice that, in order to obtain more accurate rates of cases with COVID-19, population size in 2020 by province was linearly extrapolated using the corresponding information in the population and housing censuses from 2011 and 2016." It is not my field, but I am guessing that the demographers have techniques for anticipating "future" populations that would be more robust than simple linear extrapolation.

Reviewer #2: Details in paragraph starting on L62 seem more appropriate for the discussion than an introduction for the present analysis (e.g., chronic disease, epidemiological transition). I suspect the authors intended to introduce the importance of sociodemographic factors for infectious diseases because they examine some in their analysis. A general introduction to each one (or major themes) would be preferable to explain why these sociodemographic variables were chosen among all others. 

The authors highlight spatial analyses have been applied to previous infectious diseases but only cite one investigation of malaria (Chipeta et al., 2019). More citations are necessary to support the authors’ claim. 

The aim of the paper in the sentence on L78 appears hidden among the statements of broader impact. The sentence also contains detail previously shared in the same paragraph. Clearing up the final sentence would end the Introduction on a strong foundation. 

The authors declare the data came “mainly” from two sources, which suggests there are other sources. What were the other data sources?

Variables were used raw or calculated, but it is not clear which variables fell under those two categories or how the variables were calculated. 

The authors made efforts to extrapolate population levels of each province to more contemporary levels but did not for sociodemographic or climate variables, why not? If outside the scope of this paper, the authors can include a discussion of the potential limitations in the data availability. 

The authors are missing citation(s) for why a linear extrapolation for population was used. Instead of extrapolating, authors could consider using predicted data from global models such as WorldPop (doi: 10.5258/SOTON/WP00645) and spatially aggregating by province. 

How was TEI standardized?

The average temperature of provincial capital cities and annual precipitation levels in 2015 are climactic variables and seem out of place among the sociodemographic factors in the analysis. There was no reference to this type of variable in the Introduction. The choice of a single temperature of the capital city of each province appears arbitrary without supporting detail from the authors. Were annual precipitation levels also from the capital cities? If a single metric is desired, an average temperature or total precipitation across an entire province would be more representative of climate than a single city. 

Suggestion to convert the number of physicians and hospital beds to a rate to account for population levels. 

Suggestion to use “cumulative” cases instead of “total number” of cases and how were cases confirmed? What is the official Iranian definition?

For the spatial linear models, the authors examined log-transformed cumulative case frequency instead of rates without explanation, why? 

The authors did not assess possible multicollinearity of the sociodemographic (and climate) variables before conducting their spatial linear models. It is also not clear in Methods if the authors conducted univariate spatial linear models or multivariate spatial linear models.

Reviewer #3: Need a bit of clarification of the methods.

**Results**

-Does the analysis presented match the analysis plan?

-Are the results clearly and completely presented?

-Are the figures (Tables, Images) of sufficient quality for clarity?

Reviewer #1: (No Response)

Reviewer #2: General: Keep a consistent font, spacing, and number of significant figures for reported stats. Also, consider covering “multiplicative effects” to a percent increase.

The Results contain a large amount of results interpretation especially in subsections entitled “Predictors of COVID-19 spatial spread in Iranian provinces” and “Spatial lag predictors and province clusters.” All interpretations should be moved to the Discussion. 

The sentence on L152 reads awkwardly.

I see a light blue color rather than purple in Figure 2A.

The syntax L167-168 suggests an interpretation of results and not a reporting of results.

Details on L188-190 are methodological and should be moved to the Methods.

Consider using a different word for “confirmed” on L199 with a neutral verb. 

The global spatial autocorrelation between cases and temperature is insignificant by your specified threshold (L204), and should be interpreted cautiously. 

Consider removing “as expected” on L213 from Results.

Reviewer #3: Yes

**Conclusions**

-Are the conclusions supported by the data presented?

-Are the limitations of analysis clearly described?

-Do the authors discuss how these data can be helpful to advance our understanding of the topic under study?

-Is public health relevance addressed?

Reviewer #1: (No Response)

Reviewer #2: The section could start on a stronger foundation if the title sentence ended before the clause “probably due to...” and save the second clause as a possible explanation of the spatial heterogeneity (with supporting citations). 

The authors’ focus on the sociodemographic findings and their broader impact (as well as using words such as “determinants” or “leads to”) should be tempered because their choice in methodology was exploratory. For example, bivariate analyses are useful to explore the relationship between case rate and a spatial variable, but they do not control the correlation between the two at each location and should be interpreted with caution. A discussion of this limitation is missing and can be combined with L181-185 in the Discussion.

Even though the authors introduced the possibility of underreporting of COVID-19 cases in Iran in the Introduction (L60-61), the authors do not discuss reporting as a limitation in their paper. Regardless of the reliability of data reported in the Statistics Center of Iran, the degree of testing or availability of testing and their possible spatial heterogeneity was not accounted for or discussed in the paper as a limitation to their analysis.

Using “protective effect” suggests causation and should be reworded in the context of this exploratory analysis. Also, recommending public health action based on univariate associations should be provided with caution because possible confounding by other factors has not been accounted for in the present analyses.

Also missing is a discussion of the similarity in raw and smoothed rates as well as a discussion of of the differences in the spatially lagged and spatial error models.

The strengths of the study (L276) focus on the broader impacts and do not comment on the strength of the methodology and overall approach. 

The authors did not discuss potential limitations of their climate analysis. See details in previous comment in Methods.

On L287, I suspect the authors mean using projected and reported “population” not cases. Not clear how this supports study “feasibility” but rather robustness of their extrapolation approach. 

In addition to a future direction of using spatio-temporal analyses, the authors could consider a multivariate spatial regression to account for some of the limitations in their present analysis.

Reviewer #3: The discussion section needs to be rewritten to avoid language confusion correlation with causality.

**Editorial and Data Presentation Modifications?**

Reviewer #1: "Iran has one of the highest COVID-19 mortality rates fluctuating between 8 to 18 percent daily, and..." This statement is quite unclear ... 8-18% of the individuals infected, or what?

"The large count or COVID-19 cases..." should this be "of"?

"... that improves variance instability proper of rates estimated in small-sized spatial units..." I cannot understand this phrase.

Reviewer #2: Tables/Figures:

T1: An indication of which variables were calculated and the source of all data would improve the table. 

General: Colorkey titles should be descriptive instead of raw variable name. Translate “conteo” to “count”.

F1B: The highest level in the colorkey has 0 provinces. Consider removing and reorganizing the colors.

F2: Even though the colors are geoDa defaults, consider custom colors to help with colorblind readers.

Mechanics:

General typo “SARS-COV2” is “SARS-CoV-2”

L62 typo

Abstract:

The authors should report more specific details from their study including: 1) the time frame of the study, 2) the name of the spatial analysis techniques, 3) statistical results (e.g., effect size, p-value) from their analysis, 4) at least one limitation or caveat to their analysis.

Reviewer #3: Some wording, highlighted in my review

**Summary and General Comments**

Reviewer #1: The statement "Spatial statistics have emerged as a novel multidisciplinary tool for the analysis of spatial epidemiology ..." is a bit overblown. That is, spatial statistics have been around for decades, and has been the basis for spatial epidemiology. I don't see the "novel" there, although there are certainly novel spatial statistics that could be applied to these questions. I am just wanting to see no overblown statements.

More generally, I think that this manuscript presents interesting results, although perhaps it is a bit of using complex spatial statistics to show something that is more or less obvious ... COVID-19 cases in the north, and fewer in the south. I am not sure that there is a deep and profound impact of these results on the understanding of this outbreak.

Reviewer #2: The authors provide a spatial analysis of a timely topic of global proportion (COVID-19) in a country with high mortality outside of China in the early phase of the pandemic (Iran). The paper is relevant to global health and examines COVID-19 in a resource-poor nation and is, therefore, relevant to the journal. The authors suggest the largest contribution of their paper is to inform policy, instead it is my opinion the impact of this paper is methodological whereby on-going COVID-19 models must account for the spatial autocorrelation in cases, as the authors mention briefly. The ecological study attempts to identify social factors related to case rates, which are appropriately stabilized using spatial smoothing methodology. However, there are major concerns that affect the design and clarity of the study that should be considered before publication.

Reviewer #3: NA

PLOS authors have the option to publish the peer review history of their article (what does this mean?). If published, this will include your full peer review and any attached files.

Reviewer #1: No

Reviewer #2: Yes: Ian Buller, Ph.D., M.A.

Reviewer #3: No
---

## [Decision Letter · Decision Letter 1]

23 Aug 2020

Dear Dr. Ramírez-Aldana,

Thank you very much for submitting your manuscript "Spatial analysis of COVID-19 spread in Iran: Insights into geographical and structural transmission determinants at a province level" for consideration at PLOS Neglected Tropical Diseases. As with all papers reviewed by the journal, your manuscript was reviewed by members of the editorial board and by several independent reviewers. The reviewers appreciated the attention to an important topic. Based on the reviews, we are likely to accept this manuscript for publication, providing that you modify the manuscript according to the review recommendations. 

From Associate Editor Town Peterson:

It appears that the authors conducted a rather profound revision, and my reviewer, who was previously critical, is now mostly happy. They do point out, and I agree, that an analysis that includes a model selection step would be important--so I am saying "minor revision," but I would like to see this analysis added before I do a final acceptance. So please work through the reviewer's comments, and pay special attention to the model selection part? 

All the best, ATP

Sincerely,

Townsend Peterson

Associate Editor

Justin Remais

Deputy Editor

From Associate Editor Town Peterson:

It appears that the authors conducted a rather profound revision, and my reviewer, who was previously critical, is now mostly happy. They do point out, and I agree, that an analysis that includes a model selection step would be important--so I am saying "minor revision," but I would like to see this analysis added before I do a final acceptance. So please work through the reviewer's comments, and pay special attention to the model selection part? 

All the best, ATP

Reviewer's Responses to Questions

**Key Review Criteria Required for Acceptance?**

**Methods**

-Are the objectives of the study clearly articulated with a clear testable hypothesis stated?

-Is the study design appropriate to address the stated objectives?

-Is the population clearly described and appropriate for the hypothesis being tested?

-Is the sample size sufficient to ensure adequate power to address the hypothesis being tested?

-Were correct statistical analysis used to support conclusions?

-Are there concerns about ethical or regulatory requirements being met?

Reviewer #2: After the previous round of review, the authors included a test for multicollinearity and reduced the variables used in their spatial multivariate linear models. However, the present study uses all available explanatory variables simultaneously in a model without a model selection scheme, which may increase the chance their models are overfitting the data. The objective of this part of the present analysis is “to identify variables that significantly impact the number of log-transformed COVID-19 cases” performing model selection will help achieve their goal. 

In L148, the data agreement can be paraphrased and shortened, or removed.

In L368, the authors mention the possibility “that some variables have a time lagged effect over their response” but the authors do not test for temporal lag in their analysis. Why not? If it is because data was unavailable as mentioned in L367, then I recommend discussing this as a limitation of the study.

**Results**

-Does the analysis presented match the analysis plan?

-Are the results clearly and completely presented?

-Are the figures (Tables, Images) of sufficient quality for clarity?

Reviewer #2: Recommend sentence in L243 to be moved to discussion.

Recommend adding the figure of the multicollinearity matrix to supplemental materials.

**Conclusions**

-Are the conclusions supported by the data presented?

-Are the limitations of analysis clearly described?

-Do the authors discuss how these data can be helpful to advance our understanding of the topic under study?

-Is public health relevance addressed?

Reviewer #2: The relationship between climate and SARS-CoV-2 transmission is a contentious topic, one that requires more investigation. The present study lends mixed support to this relationship. And while the authors cite a review (43) and many studies within that review have found a similar relationship to one of the results within the present study, it would be important to note contrary findings. Such as, for example, Jamil et al. 2020 (https://www.medrxiv.org/content/10.1101/2020.03.29.20046706v2) who is also cited and discussed within (43), especially in light of the present study climate data limitations.

Recommend replacing “though with a p-value of 0.103” with “not statistically significant” or something similar to help the reader interpret the results.

**Editorial and Data Presentation Modifications?**

Reviewer #2: There are grammatical errors throughout the newly added sections (highlighted in yellow), I recommend light copyediting.

**Summary and General Comments**

Reviewer #2: The authors return with a greatly improved manuscript of their study in a country affected greatly early in a global pandemic. The authors responded thoroughly, thoughtfully, and professionally to the previous reviewers, evident especially in the improvements to the tables and figures. The methods are clearer and their limitations are more encompassing than the initial submission. A few methodological concerns remain regarding the investigation into variables that are associated with cases as well as caution for the interpretation of the relationship between climatic variables and cases.

PLOS authors have the option to publish the peer review history of their article (what does this mean?). If published, this will include your full peer review and any attached files.

Reviewer #2: Yes: Ian Buller, Ph.D., M.A.
---

## [Editor Report · Decision Letter 2]

12 Oct 2020

Dear Dr. Ramírez-Aldana,

We are pleased to inform you that your manuscript 'Spatial analysis of COVID-19 spread in Iran: Insights into geographical and structural transmission determinants at a province level' has been provisionally accepted for publication in PLOS Neglected Tropical Diseases.

Best regards,

Townsend Peterson

Associate Editor

Justin Remais

Deputy Editor

---

## [Editor Report · Acceptance letter]

3 Nov 2020

Dear Dr. Ramírez-Aldana,

We are delighted to inform you that your manuscript, "Spatial analysis of COVID-19 spread in Iran: Insights into geographical and structural transmission determinants at a province level," has been formally accepted for publication in PLOS Neglected Tropical Diseases.

Best regards,

Shaden Kamhawi

co-Editor-in-Chief

Paul Brindley

co-Editor-in-Chief
